# An Evaluation of the Usefulness of Selected Screening Methods in Assessing the Risk of Malnutrition in Patients with Inflammatory Bowel Disease

**DOI:** 10.3390/nu16060814

**Published:** 2024-03-13

**Authors:** Małgorzata Godala, Ewelina Gaszyńska, Konrad Walczak, Ewa Małecka-Wojciesko

**Affiliations:** 1Department of Nutrition and Epidemiology, Medical University of Lodz, 90-752 Lodz, Poland; ewelina.gaszynska@umed.lodz.pl; 2Department of Internal Medicine and Nephrodiabetology, Medical University of Lodz, 90-549 Lodz, Poland; konrad.walczak@umed.lodz.pl; 3Department of Digestive Tract Diseases, Medical University of Lodz, 90-153 Lodz, Poland; ewa.malecka-panas@umed.lodz.pl

**Keywords:** screening tools, inflammatory bowel disease, Crohn’s disease, ulcerative colitis, malnutrition

## Abstract

The aim of this study was to assess the prevalence of malnutrition risk in patients with IBD using different scales to evaluate their usefulness as first-step screening tools for the diagnosis of malnutrition using the GLIM criteria in patients with inflammatory bowel disease. This study included 82 patients with IBD. The Mini Nutritional Assessment, Malnutrition Universal Screening Tool, Saskatchewan IBD-Nutrition Risk and Malnutrition Screening Tool were used to assess malnutrition risk in the study group. In order to diagnose malnutrition, the GLIM criteria were used. According to the GLIM recommendations, malnutrition was diagnosed in 60 patients with IBD (73.17%). Depending on the applied screening tools, the prevalence of moderate and/or high-risk malnutrition in patients with IBD ranged from 20.25% to 43.59%. The highest level of accuracy (ACC) was noted for the MST and MUST questionnaires (92.50% and 90%, respectively), followed by the SASKIBD-NR test (89.97%) and the MNA questionnaire (83.33%). The results of our study indicate a high prevalence of malnutrition in patients with IBD. Thus, there is a need to conduct routine assessments of malnutrition risk using validated scales. The MUST scale seems promising in the assessment of malnutrition risk in patients with IBD as a first step in the assessment of malnutrition using the GLIM criteria.

## 1. Introduction

The inflammatory bowel disease (IBD) group includes ulcerative colitis (UC) and Crohn’s disease (CD). Epidemiological data indicate that the incidence of IBD is increasing worldwide. Over the past century, there has been a significant increase in the general prevalence of IBD, particularly in children. An increase in the prevalence of IBD in developing countries has also been observed over the past 50 years [1,2]. According to recent Polish epidemiological data, the prevalence of IBD is estimated to be almost 100,000 people with a peak in those between 20 and 40 years of age [3,4]. The aetiopathogenesis of IBD is not fully elucidated. It is currently believed that IBD involves immunological disturbances activated by environmental factors in genetically predisposed individuals [3,4].

According to ESPEN (European Society for Parenteral and Enteral Nutrition) data, patients with inflammatory bowel disease are at particular risk of malnutrition due to inadequate energy supply, the introduction of elimination diets and the occurrence of diarrhoea [5,6,7,8,9,10,11,12]. Systematic assessments of the nutritional statuses of patients with IBD are recommended as malnutrition worsens the prognosis and quality of life and increases complication rates and mortality [9,10,13]. Anthropometric methods of assessing nutritional status, such as body composition analyses or BMI (body mass index) calculations may be insufficient in patients with IBD because the measurements are taken at one time point, whereas detecting changes in these parameters is more informative. Additionally, they do not assess energy and nutrient requirements. Additionally, they do not take into account parameters related to disease activity such as significant weight loss, clinical symptoms (diarrhoea and abdominal pain) or inflammation. In order to fully assess the prevalence of malnutrition in patients with IBD, questionnaires are used to allow the instant and early diagnosis of those at risk of malnutrition or who are already malnourished.

The lack of consensus on the diagnosis of malnutrition not only makes it difficult to compare the prevalence of malnutrition, but also causes discrepancies in formulating recommendations and monitoring results. In order to standardise criteria and enable to carry out analyses of malnutrition in different regions of the world, the GLIM (Global Leadership Initiative on Malnutrition) initiative was established. The GLIM was created by the ESPEN (European Society for Parenteral and Enteral Nutrition), ASPEN (American Society for Parenteral and Enteral Nutrition), PENSA (Parenteral and Enteral Nutrition of Asia) and FELANPE (Federation Latinoamericana de Terapia Nutricional, Nutricion Clinica, y Metabolismo). The results of the work undertaken by the GLIM in 2016 resulted in a formulation of new criteria for the diagnosis of malnutrition [14,15]. GLIM experts prepared two classes of assessment criteria—phenotypic criteria, used to determine the severity of malnutrition, and aetiological criteria, which are considered crucial in setting standards for intervention. The criteria were developed on the basis of parameters assessed with malnutrition assessment tools that are commonly applied in different countries around the world. In order to diagnose malnutrition, we need at least one phenotypic and one aetiological criterion. Diagnosing malnutrition with the GLIM criteria allows us to assess its severity and implement appropriate nutritional management. The first step recommended by GLIM experts in the diagnosis of malnutrition is to identify patients who are at increased risk of malnutrition with the use of any validated screening tools [14].

The Nutritional Risk Screening 2002 (NRS-2002), Malnutrition Universal Screening Tool (MUST), Malnutrition Screening Tool (MST) and IBD-specific tests, such as the Malnutrition Inflammation Risk Tool (MIRT) and Saskatchewan IBD-Nutrition Risk (SASKIBD-NR), are the most common tools used in the detection of malnutrition [9,16,17]. A meta-analysis of data on the use of questionnaire methods for assessing malnutrition in patients with IBD, conducted in 2019, showed discrepancies between the used screening tools. The authors of this paper emphasised the need for further research assessing the usefulness of questionnaire methods and their validation [10]. A need to formulate new screening tests taking phenotypic features and aetiological parameters into account was also postulated [16,17,18,19]. 

The aim of this study is to assess the prevalence of malnutrition risk in patients with IBD using the MNA (Mini Nutritional Assessment), MUST, MST and SASKIBD-NR scales and to evaluate their usefulness as first-step screening tools for the diagnosis of malnutrition using the GLIM criteria.

## 2. Materials and Methods

### 2.1. The Characteristics of the Study Group

This retrospective study included 82 non-Asian patients with IBD, including 48 patients with CD and 34 patients with UC, who were admitted to the Department of Digestive Tract Diseases of the Medical University of Lodz. Exclusion criteria for the study were ongoing cancer and metabolic and cardiovascular diseases. The disease activity was assessed using the CDAI (the Crohn’s disease activity index) in patients with CD and the Partial Mayo Score in patients with UC [20,21]. Clinical and endoscopy remission and no rectal bleeding, as well as a normal or quiescent appearance of the mucosa at endoscopy, were defined as normal stool frequency in the past 6 months. 

All patients had their heights and weights measured to determine BMI. Undernutrition was diagnosed for BMI < 18.5 kg/m^2^, normal weight was assigned for BMI between 18.5 and 24.9 kg/m^2^, overweight was assigned for BMI between 25.0 and 29.9 kg/m^2^ and obesity was assigned for BMI over 30.0 kg/m^2^. In addition, body composition was measured using the electrical bioimpedance method with an InBody 270 apparatus. Based on the results, the level of fat-free mass (FFM) was determined, and the fat-free mass index (FFMI) was calculated by dividing the FFM expressed in kg by height in m^2^ [14].

Food intake was assessed using a 24 h questionnaire interview, performed three times in each subject. The mean energy intake was assessed using the computer programme Diet 6.0 (licence number 52/PD/2022) [22]. In addition, the frequency of consumption of selected groups of alimentary products was assessed qualitatively and quantitatively using the Food Frequency Questionnaire (FFQ) in the last month before the study. 

Blood was collected on an empty stomach from the ulnar vein in order to determine blood count and C-reactive protein (CRP) levels. 

### 2.2. GLIM Criteria

In order to diagnose malnutrition, at least one phenotypic and one aetiological criterion are required. Phenotypic criteria include unintentional weight loss (more than 5% in the past 6 months or more than 10% in a period longer than 6 months), low body mass index (BMI < 20 kg/m^2^ up to 70 years of age and <22 kg/m^2^ in the elderly) and low muscle mass (FFMI < 17 kg/m^2^ in men and <16 kg/m^2^ in women), whereas aetiological criteria are reduced food intake or reduced absorption (up to no more than 50% of the energy value of the diet in more than a week, or any reduction in more than two weeks, or any changes in the gastrointestinal tract that reduce absorption), burden of disease or inflammation (inflammation or chronic disease contributing to increased energy expenditure and increased muscle catabolism). There are two stages of malnutrition according to the GLIM—moderate (stage 1) and advanced (stage 2). Advanced malnutrition is diagnosed on the basis of weight loss which is higher than 10% and lasts for a period of 6 months, on the basis of weight loss which is higher than 20% and lasts for longer than 6 months or a BMI lower than 18.5 kg/m^2^ in people under 70 years of age or lower than 20 kg/m^2^ in older people [14].

### 2.3. Screening Tests

#### 2.3.1. MNA

The questionnaire consists of a total of 18 questions. The anthropometric assessment section includes a measurement of the mid-arm circumference and calf circumference as well as the BMI, and the risk of malnutrition in this scale is diagnosed for a BMI below 23 kg/m^2^. The MNA scale includes an assessment of the degree of weight loss over a 3-month period, elements of dietary history and other factors that increase the risk of malnutrition, such as living away from home, taking more than three medications per day, reduced mobility and independence, psychological stress, neuropsychological disorders, epidermal abrasions and/or ulcers. The MNA questionnaire also includes a self-assessment of nutritional and health status. The questionnaire is designed to score patients’ malnutrition. A score of 24–30 points indicates a risk of malnutrition, and 17–23.5 points indicate a moderate risk of malnutrition, while a score of <17 points indicates a very high risk of malnutrition or the presence of malnutrition [23].

#### 2.3.2. MUST

The MUST scale assesses the risk of malnutrition on the basis of the BMI (>20 kg/m^2^—0 points; BMI between 18.5 and 20.0 kg/m^2^—1 point; BMI < 18.5 kg/m^2^—2 points), unintentional weight loss in the past 3–6 months (<5%—0 points; 5–10%—1 point; >10%—2 points) and inability to eat for more than 5 days, e.g., critical, surgical or neurological patient (2 points). A score of 0 points indicates a low risk of malnutrition, 1 point indicates a moderate risk of malnutrition, and a high risk of malnutrition is diagnosed for a score ≥ 2 points [10]. 

#### 2.3.3. SASKIBD-NR

The SASKIBD-NR scale assesses the risk of malnutrition manifested by clinical symptoms (vomiting, nausea, diarrhoea and lack of appetite) for longer than 2 weeks (no symptoms—0 points; 1–2 symptoms—1 point; ≥3 symptoms—2 points); insufficient energy intake due to lack of appetite (no—0 points; yes—2 points); restricting intake of any group of alimentary products (no—0 points; yes—2 points); unintentional weight loss in the past month (no—0 points; possibly—1 point; yes—2.3–4.5 kg—1 point; 4.5–7 kg—2 points; >7 kg—3 points). The malnutrition risk was considered high for a score ≥ 5 points [17].

#### 2.3.4. MST

The MST scale assesses the risk of malnutrition based on unintentional weight loss over the past month (1–5 kg—1 point; 6–10 kg—2 points; 5–11 kg—3 points; >15 kg—4 points; possibly—2 points) and reduced dietary energy intake due to lack of appetite (yes—1 point). A score of 0–1 points indicates a low risk of malnutrition, whereas a score of ≥ 2 points indicates a high risk [24]. 

### 2.4. Statistical Analysis

Data were expressed as the mean and standard deviation for quantitative variables and as integers and percentages for qualitative variables. Normality of distribution was assessed using the Shapiro–Wilk test. For univariate analysis, the Mann–Whitney U test was used when the grouping variable was dichotomous. The Kruskal–Wallis H-test was performed when the grouping variable had more than two categories. Concordance between GLIM criteria for malnutrition and the four screening tools was assessed using sensitivity, specificity, the Positive Likelihood Ratio, Negative Likelihood Ratio, Accuracy and Youden’s Index J for each scale. ROC curves were plotted to assess the concordance of the screening tools with the diagnosis of malnutrition by BMI.

The study was conducted in accordance with the Declaration of Helsinki and approved by the Bioethics Committee of the Medical University of Lodz (No. RNN/70/22/KE). All of the respondents gave written consent to participate in the study. 

## 3. Results

### Characteristics of the Study Group

This study included 48 patients with CD and 34 patients with UC. A total of 17 patients with CD (35.42%) and 17 patients with UC (50%) were in clinical and endoscopic remission. Patients with CD and UC did not differ significantly in terms of disease duration and gender structure. However, patients with CD were younger than patients with UC (34.66 ± 10.24 years vs. 43.06 ± 11.87 years, *p* = 0.001), and more of them had undergone bowel resections in the past (35.42% vs. 14.71%, *p* = 0.0315) (Table 1).

Of all patients, 6 patients with CD (12.5%) and 4 patients with UC (11.76%) were underweight based on BMI; 24 patients with CD (50%) and 14 patients with UC (41.18%) were normal-weight individuals, and 18 patients with CD (37.5%) and 16 patients with UC (47.06%) were overweight or obese. Unintentional weight loss over 3–6 months was reported in 50 respondents (60.96%); 38 respondents (46.34%) demonstrated a weight loss of more than 10%.

With regard to the FFM, the mean FFM level in the patients with IBD was 75.14 ± 11.23% and did not differ significantly between the patients with CD and UC. There were also no significant differences between the patients with CD and UC in terms of the FFMI. Its mean value was 17.97 ± 3.26 kg/m^2^ in the entire study group. In 18 patients with IBD (21.85%), the FFMI was reported to be too low.

The mean energy intake of patients with IBD was 1575 ± 302 kcal per day and did not differ significantly between patients with CD and UC. Of all patients, 30 (36.59%) did not meet 50% of the recommendations for dietary energy intake. It regarded patients with CD significantly more often than patients with UC (41.67% vs. 29.41%, respectively, *p* = 0.0321). The majority of patients (*n* = 72, 87.8%) declared the elimination of at least one group of alimentary products.

According to the GLIM recommendations, malnutrition was diagnosed in 60 patients with IBD (73.17%), including 36 patients with CD (75%) and 24 patients with UC (70.59%). Among patients diagnosed with malnourishment, 50 (83.33%) had unintentional weight loss, 15 (25%) had a low body mass index and 18 (30%) had a low muscle mass. All patients with IBD and malnourishment were characterised by reduced food intake or food group eliminations. The first degree of malnutrition was diagnosed in 38 patients with IBD (46.34%), including 23 patients with CD (47.92%) and 15 patients with UC (44.12%). The second degree of malnutrition according to the GLIM criteria was diagnosed in 22 patients with IBD (27.5%), including 13 patients with CD (27.66%) and 9 patients with UC (27.27%). The second degree of malnutrition was found significantly more often in patients with active disease compared to patients in remission (45.24% vs. 7.89%, respectively, *p* = 0.0005) (Table 2). 

Depending on the applied screening tools, the prevalence of moderate and/or high-risk malnutrition in patients with IBD ranged from 20.25% to 43.59%. The MNA and MUST questionnaires revealed the highest proportion of patients at moderate and high risk of malnutrition (43.59% and 37.5%, respectively), while the MST and SASKIBD-NR questionnaires showed the lowest risk (25% and 20.25%, respectively). Regardless of the used method, the disease type and occurrence of past intestinal resections did not significantly contribute to the prevalence of high-risk malnutrition. In contrast, a high risk of malnutrition was observed significantly more often in patients with exacerbated symptoms in comparison to patients in remission for all used screening tools. 

A comparison of the malnutrition risk assessment tools and GLIM criteria revealed that the MNA and MUST questionnaires were characterised with the highest sensitivity (100%) in all patients with IBD. The MST questionnaire demonstrated 81.82% sensitivity, while the SASKIBD-NR test was characterised with the lowest sensitivity among all analysed tools (68.18%) (Table 3).

A comparison of the malnutrition risk assessment tools and GLIM criteria showed that the SASKIBR-NR test and MST questionnaire demonstrated the highest specificity (98.25% and 96.55%, respectively). The MUST questionnaire was characterised with 86.21% specificity. In contrast, the specificity level observed for the MNA questionnaire was 77.19%.

The highest level of accuracy (ACC) was noted for the MST and MUST questionnaires (92.50% and 90%, respectively), followed by the SASKIBD-NR test (89.97%) and the MNA questionnaire (83.33%).

ROC curves were plotted to compare the applied malnutrition risk assessment tools with BMI. The largest area under the ROC curve was noted for the SASKIBD-NR test criteria (AUC = 0.9916; 95% CI: 0.9779–1.0000; *p* < 0.0001), followed by the MNA (AUC = 0.8887; 95% CI: 0.8164–0.9610; *p* < 0.0001), MUST (AUC = 0.8603; 95% CI: 0.7768–0.9439; *p* < 0.0001) and MST (AUC = 0.8212; 95% CI: 0.6988–0.9437; *p* < 0.0001) (Figure 1).

## 4. Discussion

All persons at increased risk of malnutrition should undergo screening tests [25,26,27,28,29]. GLIM experts did not explicitly recommend a specific tool for screening purposes but suggested that any of the validated tools could be used, e.g., the NRS 2002 (Nutritional Risk Score), SGA (Subjective Global Assessment), MUST or MNA [30,31,32,33,34,35,36]. According to Polish legislation, all hospitalised individuals, except for those treated in hospital emergency departments, are subject to the screening procedure. Adult patients are assessed with the application of NRS 2002 or SGA tools [25]. 

Patients with IBD undoubtedly belong to a group of patients who are at increased risk of malnutrition, which, if diagnosed early, can result in efficient treatment and improved quality of life [19,37]. However, the selection of proper criteria for assessing malnutrition is crucial but still needs to be discussed, as this would enable us to accurately identify those who are at increased risk of malnutrition. A number of nutritional screening tools are available, and the MUST, MIRT, NRI, NRS2002, SGA and MNA questionnaires, among others, were described in reports on in-patients or out-patients with IBD [14,38]. These screening tools for the assessment of malnutrition risk contain similar criteria. They all use the body mass index (BMI) or body composition, unintentional weight loss in the past 3 months (except NRI) and disease activity, assessed by the presence of clinical symptoms or the result of biochemical tests, such as albumin (NRI) or CRP (MIRT). Among other screening elements, a change in food intake was included (MUST, MNA and NRS2002), and according to the MNA criteria, mobility limitations and the presence of depression or dementia are likely to be associated with a higher risk of malnutrition [14,39,40].

In our study, we assessed the prevalence of malnutrition in patients with IBD based on the latest GLIM expert recommendations. According to these guidelines, we found malnutrition in 73.17% of all patients, including second-degree malnutrition in 27.5% of the respondents. In a study by Huang et al., conducted in a group of 73 patients with IBD, the prevalence of malnutrition assessed using GLIM criteria was 58.9%, and 30.14% of the respondents were affected by second-degree malnutrition [41]. Zhang et al. conducted a study on 238 hospitalised patients with IBD and revealed that malnutrition according to the GLIM criteria was diagnosed in 60.1% of them [18]. In contrast, in a study by Fiorindi et al., conducted on a group of patients with IBD who qualified for surgery, malnutrition according to the GLIM criteria was reported in 40% of the patients [42]. The lower prevalence of malnutrition in this study may result from the fact that a normal nutritional status is a prerequisite before a planned surgery. 

In the study group, we also used questionnaires to screen for malnutrition risk. After using the MNA, MUST, MST and SASKIBD-NR questionnaires, we found a combined moderate and high risk of malnutrition in 43.59%, 37.5%, 25% and 20.25% of patients, respectively.

The MNA questionnaire is one of the validated methods for assessing malnutrition and is frequently used among elderly and hospitalised patients. Studies revealed malnutrition rates ranging from 31% to 47% in hospitalised and elderly patients [26,27,28,29,39,43]. There are few studies that have assessed the risk of malnutrition in patients with IBD using MNA, and their results are similar to those obtained in our own study. In a study by Pieczyńska et al., conducted in a group of 162 patients with IBD, a high risk of malnutrition according to the MNA criteria was found in 53.1% of the respondents [44]. In a study by Presch et al., a risk of malnutrition or the presence of malnutrition was demonstrated in 62% of patients with IBD [45]. 

The MUST questionnaire is the second validated malnutrition risk assessment questionnaire used in this study. It is often applied in routine assessments of malnutrition risk in hospitalised patients. There are also studies assessing the risk of malnutrition in patients with IBD using this tool. In a study by Haskey et al., a high risk of malnutrition according to the MUST criteria was diagnosed in 15.5% of patients with IBD [17]. Fiorindi et al. assessed the risk of malnutrition twice in a group of patients with IBD. The MUST questionnaire indicated a high risk of malnutrition in 26% and 28% of patients [42,46]. Einav et al. noted a 36.4% prevalence of malnutrition risk in patients with IBD in whom the MUST questionnaire was applied [47]. Csontos et al. used the MUST questionnaire in their study and observed that 31.8% of the respondents in a group of 173 patients with IBD demonstrated a moderate to high risk of malnutrition [48]. 

The MST questionnaire is used as a validated screening tool to assess the malnutrition risks of hospitalised patients in many countries. The Nutrition Day project, initiated by the Medical University of Vienna and the ESPEN, assessed the prevalence of malnutrition among hospitalised patients in 25 European countries, including Poland, using the above questionnaire. It showed a high risk of malnutrition, i.e., 29.9% in all participating centres, whereas among Polish hospitalised patients, malnutrition was found in 24.5% of the respondents [49]. Few studies assessing the risk of malnutrition with the use of the MST questionnaire in patients with IBD are also available. In a study by Fiorindi et al., 28% of patients with IBD who qualified for a surgery demonstrated a high risk of malnutrition [46]. In another study by Fiorindi et al., 26% of patients with IBD exhibited a high risk of malnutrition [42]. 

In this study, we also used the SASKIBD-NR questionnaire designed by Canadian researchers. The questionnaire was developed in response to the lack of a screening tool which could be used to assess the risk of malnutrition and which includes key risk factors for patients with IBD. Patients with CD and UC frequently follow self-imposed dietary restrictions and thus demonstrate a higher risk of nutrient deficiencies. Most available screening tools assess the risk of malnutrition on the basis of BMI results. Additionally, studies have confirmed that the BMI levels of patients with IBD in remission regarding disease symptoms are similar to the ones found in healthy and well-nourished individuals [38,50,51,52,53,54,55]. A study by Haskey et al. found a moderate to high risk of malnutrition in 19.1% of ambulatory patients with IBD [17]. In a study by Fiorindi et al., who applied the SASKIBD-NR criteria, a moderate or high risk of malnutrition was found in 40% of patients with IBD [42]. In a study conducted by Taylor et al. on 245 patients with IBD, a moderate or high risk of malnutrition according to the SASKIBD-NR criteria was reported in 36% of patients [56]. 

Both the SASKIBD-NR and MST questionnaires were characterized by high specificity compared to the GLIM criteria. But regarding the concurrence of the applied methods, the highest accuracy, which is the most important measure, was obtained for the MUST questionnaire. Similar data were obtained in a study by Zhou et al. conducted on a group of patients with gastrointestinal cancers. In this study, the highest concurrence with the GLIM criteria for malnutrition was obtained for the MUST questionnaire [39]. Also, Taylor et al. conducted a study on a group of patients with IBD and confirmed that the MUST scale was highly effective in detecting malnourished patients [56]. 

Different data were obtained in a Canadian study by Haskey et al., in which the MUST questionnaire showed insufficient sensitivity and specificity against the current criteria for malnutrition assessment in patients with IBD [17]. Additionally, a study by Csontos et al. indicated that the MUST questionnaire appeared to be insufficient to identify patients with IBD who are at risk of malnutrition. The authors found that nearly 10% of the study respondents were classified by the MUST questionnaire as patients with a low risk of malnutrition, whereas these patients exhibited worryingly low levels of muscle mass, which indicated malnutrition [48]. The different results of the studies may be associated with differences in the nutritional statuses of the patients. In both of the quoted studies, the BMI levels of the patients were higher, and this probably resulted in a misestimation of their malnutrition risks.

Selecting an appropriate screening tool to identify patients who are at risk of malnutrition is crucial as it enables appropriate therapy and dietary management methods to be implemented. The fact that available nutrition screening tools are designed to identify people who are at risk of malnutrition who have a low BMI and/or acute weight changes is its main limitation. Researchers’ opinions on the use of the BMI as an indicator of malnutrition are various. Studies have shown that the BMI is not an accurate indicator of malnutrition in patients with chronic inflammatory conditions, and that the loss of lean body mass can occur in people with normal BMI levels or even in patients who are overweight [38,50,51,57]. Given the fact that overweight and obesity are increasingly prevalent in the European population, the proportion of patients with IBD who are overweight is also increasing. On the other hand, the BMI is a simple indicator, widely known and repeatedly analysed in studies, which supports its clinical usefulness. However, by relying solely on the BMI, we may underestimate patients who are malnourished due to an inappropriate diet but whose BMI levels are normal or even high. These patients frequently used elimination diets to control disease symptoms and maintain clinical remission [58,59]. Meanwhile, avoiding certain alimentary products and/or groups of alimentary products increases the risk of micronutrient deficiencies, even if the patients’ energy intake is sufficient and they maintain a normal or even increased BMI. Thus, screening tools for malnutrition risk that include criteria other than the BMI may help identify more people who are at risk of malnutrition.

In this study, we compared screening methods for malnutrition assessment with the BMI and observed the highest efficiency for the SASKIBD-NR test. It is interesting that the authors, being critical of the BMI as a marker of malnutrition, proposed a scale without taking this indicator into account. Of the questionnaires used in this study, only the MUST and MNA questionnaires include the BMI among other risk factors, and values of 20 kg/m^2^ and 23 kg/m^2^, respectively, are considered critical. The determination of BMI cut-off points for patients with IBD still needs to be discussed. This seems to be crucial in order to adequately identify those who are at risk of malnutrition or already malnourished. The MNA, MUST and MST questionnaires were designed primarily for older and hospitalised patients. If we want to use them for patients with IBD, we probably have to establish acceptable BMI values depending on disease activity, treatment modality (outpatient/hospitalisation), existing complications, patient age or pre-disease nutritional status. The present study showed that the best results were obtained for a cut-off point of 23.8 kg/m^2^, while for the MNA and MUST questionnaires, the cut-off point value was 20.8 kg/m^2^. 

In the present study, we demonstrated a need to assess the risk of malnutrition in patients with IBD in order to develop a tool that takes into account the risk factors that are relevant in this group of patients. Of all the analysed scales, the best results were obtained for the MUST questionnaire. This tool appeared to show the highest accuracy in relation to the currently valid and recommended GLIM criteria as well as a relatively high concordance with BMI. 

The retrospective nature of this study is another limitation. It is burdened with gaps in the memory of the respondents. Furthermore, it is a single-centre study. Thus, it is conducted using a relatively small group of patients. 

## 5. Conclusions

The results of our study indicate a high prevalence of malnutrition in patients with IBD. Thus, there is a need to conduct routine assessments of malnutrition risk using validated scales. The MUST scale seems promising in the assessment of malnutrition risk in patients with IBD as a first step in the assessment of malnutrition using the GLIM criteria. However, this needs to be confirmed by further research.

## Figures and Tables

**Figure 1 nutrients-16-00814-f001:**
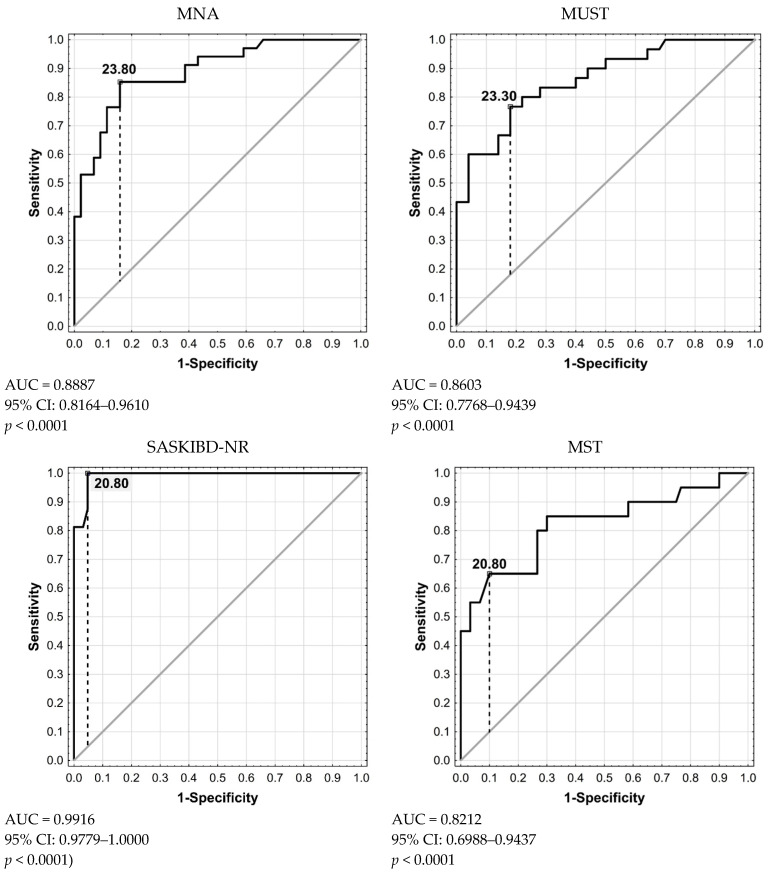
Malnutrition based on BMI versus malnutrition based on Mini Nutritional Assessment (MNA), Malnutrition Universal Screening Tool (MUST), Saskatchewan IBD–Nutrition Risk (SASKIBD-NR) and Malnutrition Screening Tool (MST) criteria; Receiver Operating Characteristic.

**Table 1 nutrients-16-00814-t001:** Baseline characteristics of patients with IBD.

	IBD *n* (%), Mean ± SD	CD *n* (%), Mean ± SD	UC *n* (%), Mean ± SD	*p*
Patients	82	48 (58.5)	34 (41.5)	
Age [years]	38.14 ± 11.65	34.66 ± 10.24	43.06 ± 11.87	0.0010
Female	42 (51.22)	28 (58.33)	14 (41.18)	0.1257
Duration of the disease [years]	8.39 ± 5.71	8.08 ± 5.87	8.82 ± 4.56	0.7234
Past intestinal partial resection	22 (26.83)	17 (35.42)	5 (14.71)	0.0315
Disease activity				
CDAI [0/1/2/3]		17(35.4)/10(20.9)/17(35.4)/4 (8.3)		
Partial Mayo Score (0/1/2/3)			17 (50.0)/0 (0)/11 (32.4)/6 (17.6)	
BMI [kg/m^2^]	24.25 ± 4.76	23.81 ± 4.88	24.81 ± 4.59	0.4705
≤18.5	10 (12.20)	6 (12.5)	4 (11.76)	0.6745
18.5–24.9	38 (46.34)	24 (50)	14 (41.18)	
≥25.0	34 (41.46)	18 (37.5)	16 (47.06)	
UWL [% body weight]	16.50 ± 8.23	16.20 ± 7.85	16.95 ± 8.97	0.5812
≤5%	6 (7.32)	4 (8.33)	2 (5.88)	0.8640
5–10%	6 (7.32)	4 (8.33)	2 (5.88)	
≥10%	38 (46.34)	22 (45.83)	16 (47.06)	
FFM %	75.14 ± 11.23	74.55 ± 12.6	75.98 ± 9.72	0.7896
FFM % (M)	80.08 ± 11.54	78.96 ± 14.38	81.20 ± 7.98	<0.0001
FFM % (F)	70.44 ± 8.73	71.39 ± 9.55	68.54 ± 6.71	<0.0001
FFMI kg/m^2^	17.97 ± 3.26	17.51 ± 3.43	18.62 ± 2.95	0.6596
FFMI kg/m^2^(M)	19.77 ± 3.26	19.32 ± 4.02	20.21 ± 2.29	<0.0001
FFMI kg/m^2^ (F)	16.26 ± 2.19	15.22 ± 2.22	16.34 ± 2.21	<0.0001
FFMI < 17 (M) or <16 (F)	18 (21.95)	12 (25.00)	6 (17.65)	0.4281
Food group elimination	72 (87.8)	44 (91.67)	28 (82.35)	0.5743
Energy intake [kcal]	1575 ± 302	1540 ± 282	1626 ± 320	0.7658
<50% energy requirements	30 (36.59)	20 (41.67)	10 (29.41)	0.0321

Body Mass Index (BMI); Unintended Weight Loss (UWL); Free Fat Mass (FFM); Free Fat Mass Index (FFMI). Assessment of malnutrition according to GLIM criteria and risk of malnutrition assessed with screening methods.

**Table 2 nutrients-16-00814-t002:** Prevalence of malnutrition diagnosis and moderate/high nutritional risk in patients with IBD, CD and UC.

	IBD *n* (%)	CD *n* (%)	UC *n* (%)	Active *n* (%)	Remission *n* (%)	*p *^,^***
**Malnutrition diagnosis**						
GLIM stage 1	38 (46.34)	23 (47.92)	15 (44.12)	24 (50)	14 (41.18)	0.9696 0.8745
GLIM stage 2	22 (27.5)	13 (27.66)	9 (27.27)	19 (45.24)	3 (7.89)	0.9853 0.0005
**Nutritional screening tools**						
MNA	34 (43.59)	22 (48.89)	12 (36.36)	24 (60)	10 (26.32)	0.2704 0.0027
MUST	30 (37.5)	17 (36.17)	13 (39.39)	23 (54.76)	7 (18.42)	0.7694 0.0008
SASKIBD-NR	16 (20.25)	10 (21.28)	6 (18.75)	13 (30.95)	3 (8.11)	0.7838 0.0250
MST	20 (25)	14 (29.79)	6 (18.18)	16 (38.10)	4 (10.53)	0.2380 0.0097

Inflammatory bowel disease (IBD); Crohn’s disease (CD); ulcerative colitis (UC); Global Leadership Initiative on Malnutrition (GLIM); Mini Nutritional Assessment (MNA); Malnutrition Universal Screening Tool (MUST); Saskatchewan IBD–Nutrition Risk (SaskIBD-NR); Malnutrition Screening Tool (MST). * *p* CD vs. UC; ** *p* active vs. remission.

**Table 3 nutrients-16-00814-t003:** Comparison of prevalence of moderate/high nutritional risk and malnutrition diagnosed by GLIM in patients with IBD.

	Value	95% CI
MNA		
Sensitivity [%]	100	83.89–100
Specificity [%]	77.19	64.16–87.26
Positive Likelihood Ratio	4.38	2.72–7.07
Negative Likelihood Ratio	0	
Positive Predictive Value [%]	61.76	50.05–72.26
Negative Predictive Value [%]	100	91.96–100
Accuracy [%]	83.33	73.19–90.82
Youden’s Index J	0.77	
MUST		
Sensitivity [%]	100	84.56–100
Specificity [%]	86.21	74.62–93.85
Positive Likelihood Ratio	7.25	3.81–13.80
Negative Likelihood Ratio	0	
Positive Predictive Value [%]	73.33	59.10–83.96
Negative Predictive Value [%]	100	92.89–100
Accuracy [%]	90	81.24–95.58
Youden’s Index J	0.86	
SASKIBD-NR		
Sensitivity [%]	68.18	45.13–86.14
Specificity [%]	98.25	90.61–99.96
Positive Likelihood Ratio	38.86	5.45–276.90
Negative Likelihood Ratio	0.32	0.18–0.60
Positive Predictive Value [%]	93.75	67.80–99.07
Negative Predictive Value [%]	88.89	81.26–93.66
Accuracy [%]	89.87	81.02–95.53
Youden’s Index J	0.66	
MST		
Sensitivity [%]	81.82	59.72–94.81
Specificity [%]	96.55	88.09–99.58
Positive Likelihood Ratio	23.73	5.99–93.94
Negative Likelihood Ratio	0.19	0.08–0.46
Positive Predictive Value [%]	90	69.45–97.27
Negative Predictive Value [%]	93.33	85.21–97.14
Accuracy [%]	92.5	84.39–97.20
Youden’s Index J	0.78	

Mini Nutritional Assessment (MNA); Malnutrition Universal Screening Tool (MUST); Saskatchewan IBD–Nutrition Risk (SASKIBD-NR); Malnutrition Screening Tool (MST).

## Data Availability

The data presented in this study are available upon request from the corresponding author. The data are not publicly available due to privacy.

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
