# Peer review of "An Evaluation of the Usefulness of Selected Screening Methods in Assessing the Risk of Malnutrition in Patients with Inflammatory Bowel Disease"

_nutrients, 2024, doi:10.3390/nu16060814_

Round 1

Reviewer 1 Report

Comments and Suggestions for Authors

Godala et al compared 4 different screening tools for malnutrition in patients with inflammatory bowel disease (IBD). They demonstrated the better diagnostic accuracy for MUST and MST tools. Main comments:

1) I suggest to compare the values provided by the tools among patients with or without previous surgery, as this may be a cause of malnutrition, especially in case of extensive resection that may lead to short bowel syndrome. This point should be analyzed and discussed.

2) Authors affirmed to have evaluated disease activity by CDAI/partial Mayo, but this is absent in patients baseline features (table 1).

3) Another interesting analysis that is missing is the correlation between disease activity indexes and nutritional tools, as Authors confirmed that disease activity strongly influences state of malnutrition (as shown in table 2).

4) Malnutrition is not only undernutrition, but also obesity, a problem that is increasing in IBD (see Losurdo G et al, World J Gastroenterol 2020). Please discuss.

5) In the Discussion, an explanation to the fact that SASKIBD and MST had better specificity than MUST and MNA is necessary.

Author Response

Dear Reviewer,

Thank you very much for your time and effort to improve the article. We have changed it according to your recommendations, hoping that this version will be satisfying.

Godala et al compared 4 different screening tools for malnutrition in patients with inflammatory bowel disease (IBD). They demonstrated the better diagnostic accuracy for MUST and MST tools. Main comments:

1) I suggest to compare the values provided by the tools among patients with or without previous surgery, as this may be a cause of malnutrition, especially in case of extensive resection that may lead to short bowel syndrome. This point should be analyzed and discussed.

In our study there were no patients with short bowel syndrome. We did not find any important differences between patient with and without previous surgery, maybe because of the fact that very small part of studied group had surgical history. We added missing information.

2) Authors affirmed to have evaluated disease activity by CDAI/partial Mayo, but this is absent in patients baseline features (table 1).

We added missing information.

3) Another interesting analysis that is missing is the correlation between disease activity indexes and nutritional tools, as Authors confirmed that disease activity strongly influences state of malnutrition (as shown in table 2).

Unfortunately, we did not find important linear corelation between disease activity indexes and nutritional tools.

4) Malnutrition is not only undernutrition, but also obesity, a problem that is increasing in IBD (see Losurdo G et al, World J Gastroenterol 2020). Please discuss.

We discussed this problem in discussion section. We added the suggested article in the references list.

5) In the Discussion, an explanation to the fact that SASKIBD and MST had better specificity than MUST and MNA is necessary.

Added.

Reviewer 2 Report

Comments and Suggestions for Authors

The authors have reviewed the usefulness of screening methods in assessing the risk of malnutrition in IBD patients.

Four different screening tools were compared to Global Leadership Initiative on Malnutrition (GLIM) criteria as gold standard. Previous studies have demonstrated differences in their ability to identify malnutrition or risk in IBD patients. This study aimed to assess the prevalence of malnutrition risk and usefulness as screening tools for diagnosing malnutrition. The topic is of interest, but there is a list of inaccuracies that should be adjusted. There are some indications that the patient cohort is somewhat different from many other patient cohorts and this may affect the evaluation of the screening tools.

1.      In the beginning of the Methods section is should be stated that this was a retrospective study, this is admitted in the “limitations paragraph” in the final parts for the Discussion.

2.      A main finding was that 60 (73.2%) of IBD patients had GLIM malnutrition, which is a very high percentage. In the Discussion please adjust the statement that 73.2% is similar to other reports, the proportion appears to be higher than in most or all other studies. It would therefore be informative to present baseline data describing the GLIM phenotype of these 60 patients; how many had significant weight loss, BMI < 20 or reduced muscle mass.

3.      In the Methods section, please define clinical and endoscopic remission, since these variables are presented in the Results line 168. What was the time interval between scoring disease activity and the screening for malnutrition risk? Were patients admitted to hospital ward or outpatients? This information may also help the reader to understand why a high proportion of patients had GLIM malnutrition.

4.      The Discussion is long and tedious to digest with many paragraphs. Please attempt to distill the most important topics and shorten.

Minor

1.      There are many abbreviations in the abstract that makes it impossible to read alone, whether this is acceptable or not is up to the Editor / publisher.

2.      In lines 40-44 When arguing that body composition and BMI are insufficient to assess nutritional status it should be specified that this concerns measurements at one time point, whereas a change in these parameters are of more informative. A significant weight loss will influence BMI and the sentences in lines 40-44 seem contradictory.

3.      A mixed IBD cohort of 82 patients was examined. In line 87 it is stated that “Malnutrition was diagnosed for BMI<18.5…” This statement or definition is confusing since GLIM uses BMI < 20 inpatients < 70 years. Please rephrase and replace by “underweight” to avoid multiple definition of malnutrition. Similarly in line 88, please replace “normal nutritional status” with “normal weight”.

4.      Line 100: please rephrase “CRP protein levels”, C-reactive protein is a protein. Please define CRP in line 100 and use the abbreviation in line 272.

5.      In the methods section, please specify that the study population consisted of non-Asian participants.

6.      Statistics. Did all variables have normal distribution, whereas only non-parametric tests were used for comparisons between groups? Please comment or adjust. Data with non-normal distribution should preferably be presented as median (range or IQR).

7.      Results. One decimal seems sufficient when presenting for instance mean age and SD. Similarly, the inconsistency between none, one and two decimals should be adjusted in lines 173 and onwards, the level of precision of these variables is only modest. Related to this, the number of digits in p-values in Table 1 is also very high, please reduce 0.1257 to 0.12 etc.

8.      Line 324: it is claimed that patients with CD and UC frequently follow dietary restrictions. Do the authors mean self-imposed restrictions or given by a health care professional in Poland? It was stated that all these patients were in remission. Did some have short bowel syndrome not specified in Table 1?

9.      In table 1 it is peculiar that five patients with UC had undergone intestinal resection. Were these five patients operated by total colectomy or other procedures?

10.   In Table 3, the headers NMA and MUST have been indented and makes it difficult to read, please adjust. Left alignment of the headers would work better.

Comments on the Quality of English Language

Fine, minor adjustments as requested above. 

Author Response

Dear Reviewer,

Thank you very much for your time and effort to improve the article. We have changed it according to your recommendations, hoping that this version will be satisfying.

Four different screening tools were compared to Global Leadership Initiative on Malnutrition (GLIM) criteria as gold standard. Previous studies have demonstrated differences in their ability to identify malnutrition or risk in IBD patients. This study aimed to assess the prevalence of malnutrition risk and usefulness as screening tools for diagnosing malnutrition. The topic is of interest, but there is a list of inaccuracies that should be adjusted. There are some indications that the patient cohort is somewhat different from many other patient cohorts and this may affect the evaluation of the screening tools.

  1. In the beginning of the Methods section is should be stated that this was a retrospective study, this is admitted in the “limitations paragraph” in the final parts for the Discussion.

    We added the information about retrospective character of the study.
  2. A main finding was that 60 (73.2%) of IBD patients had GLIM malnutrition, which is a very high percentage. In the Discussion please adjust the statement that 73.2% is similar to other reports, the proportion appears to be higher than in most or all other studies. It would therefore be informative to present baseline data describing the GLIM phenotype of these 60 patients; how many had significant weight loss, BMI < 20 or reduced muscle mass.

We deleted the sentence about “similar” results. We added baseline data describing the GLIM criteria.

  1. In the Methods section, please define clinical and endoscopic remission, since these variables are presented in the Results line 168. What was the time interval between scoring disease activity and the screening for malnutrition risk? Were patients admitted to hospital ward or outpatients? This information may also help the reader to understand why a high proportion of patients had GLIM malnutrition.

    We added missing information in the methods section.
  2. The Discussion is long and tedious to digest with many paragraphs. Please attempt to distill the most important topics and shorten.

We tried to make it more concise, although it is difficult as far as the other reviewer ask to make it longer.

Minor

  1. There are many abbreviations in the abstract that makes it impossible to read alone, whether this is acceptable or not is up to the Editor / publisher.

    Corrected.
  2. In lines 40-44 When arguing that body composition and BMI are insufficient to assess nutritional status it should be specified that this concerns measurements at one time point, whereas a change in these parameters are of more informative. A significant weight loss will influence BMI and the sentences in lines 40-44 seem contradictory.

Corrected.

  1. A mixed IBD cohort of 82 patients was examined. In line 87 it is stated that “Malnutrition was diagnosed for BMI<18.5…” This statement or definition is confusing since GLIM uses BMI < 20 inpatients < 70 years. Please rephrase and replace by “underweight” to avoid multiple definition of malnutrition. Similarly in line 88, please replace “normal nutritional status” with “normal weight”.

    Corrected.
  2. Line 100: please rephrase “CRP protein levels”, C-reactive protein is a protein. Please define CRP in line 100 and use the abbreviation in line 272.

Corrected.

  1. In the methods section, please specify that the study population consisted of non-Asian participants.

We added the information about ethnic background.

  1. Statistics. Did all variables have normal distribution, whereas only non-parametric tests were used for comparisons between groups? Please comment or adjust. Data with non-normal distribution should preferably be presented as median (range or IQR).

    All statistics have normal distribution.
  2. Results. One decimal seems sufficient when presenting for instance mean age and SD. Similarly, the inconsistency between none, one and two decimals should be adjusted in lines 173 and onwards, the level of precision of these variables is only modest. Related to this, the number of digits in p-values in Table 1 is also very high, please reduce 0.1257 to 0.12 etc.

With all due respect, this is the editors’ recommendations to leave it in this form.

  1. Line 324: it is claimed that patients with CD and UC frequently follow dietary restrictions. Do the authors mean self-imposed restrictions or given by a health care professional in Poland? It was stated that all these patients were in remission. Did some have short bowel syndrome not specified in Table 1?

    We added the inforamtion about self-imposed dietary restrictions. No one has short bowel syndrome.
  2. In table 1 it is peculiar that five patients with UC had undergone intestinal resection. Were these five patients operated by total colectomy or other procedures?

    No, these 5 patients had partial resections. We added the information in Table 1.
  3. In Table 3, the headers NMA and MUST have been indented and makes it difficult to read, please adjust. Left alignment of the headers would work better.

Corrected

Round 2

Reviewer 2 Report

Comments and Suggestions for Authors

I am impressed by the accurate and rapid response from the authors and I do not have further questions.